# Molecular Ultrasound Imaging

**DOI:** 10.3390/nano10101935

**Published:** 2020-09-28

**Authors:** Gurbet Köse, Milita Darguzyte, Fabian Kiessling

**Affiliations:** 1Institute for Experimental Molecular Imaging, University Hospital Aachen, Forckenbeckstrasse 55, 52074 Aachen, Germany; gkoese@ukaachen.de (G.K.); mdarguzyte@ukaachen.de (M.D.); 2Fraunhofer MEVIS, Institute for Medical Image Computing, Forckenbeckstrasse 55, 52074 Aachen, Germany

**Keywords:** molecular ultrasound, nanobubbles, active targeting, targeted microbubbles, angiogenesis, inflammation, thrombosis, clinical translation, molecular imaging

## Abstract

In the last decade, molecular ultrasound imaging has been rapidly progressing. It has proven promising to diagnose angiogenesis, inflammation, and thrombosis, and many intravascular targets, such as VEGFR2, integrins, and selectins, have been successfully visualized in vivo. Furthermore, pre-clinical studies demonstrated that molecular ultrasound increased sensitivity and specificity in disease detection, classification, and therapy response monitoring compared to current clinically applied ultrasound technologies. Several techniques were developed to detect target-bound microbubbles comprising sensitive particle acoustic quantification (SPAQ), destruction-replenishment analysis, and dwelling time assessment. Moreover, some groups tried to assess microbubble binding by a change in their echogenicity after target binding. These techniques can be complemented by radiation force ultrasound improving target binding by pushing microbubbles to vessel walls. Two targeted microbubble formulations are already in clinical trials for tumor detection and liver lesion characterization, and further clinical scale targeted microbubbles are prepared for clinical translation. The recent enormous progress in the field of molecular ultrasound imaging is summarized in this review article by introducing the most relevant detection technologies, concepts for targeted nano- and micro-bubbles, as well as their applications to characterize various diseases. Finally, progress in clinical translation is highlighted, and roadblocks are discussed that currently slow the clinical translation.

## 1. Introduction

Ultrasound (US) imaging, also known as sonography, is used routinely in clinics for examining various organs of the body to diagnose, localize, and characterize diseases. Its advantages are low costs, real-time imaging capability, and the lack of exposure of the patient to radioactive rays. During US imaging the body is exposed to high-frequency sound waves, which are reflected by the tissues. The US probe is detecting the echoes and by calculating the amplitude and time of the reflected waves, an image is generated. Since tissue penetration decreases with increasing frequency of the sound waves but resolution increases, harmonic imaging was developed, which considers not only the center frequency of the echoes, but also their higher harmonics [1].

US contrast agents (UCA) can be used to visualize the vasculature of the tissue. The standard UCA are small gas-filled spheres, mainly referred to as microbubbles (MB), which are stabilized by a shell layer formed of lipids, proteins or polymers [2]. MB typically have a size between 1 and 10 µm. Therefore, they do not leave the vasculature. Due to the acoustic impedance difference between gas and blood, UCA can be easily distinguished from the tissue. Hereby, an increase in the signal to noise ratio is achieved.

Furthermore, decorating UCA with targeting moieties, specific disease markers can be detected and quantified. Various intravascular targets have been proposed for MB targeting such as integrins, selectins, and cell adhesion molecules. The preclinical results indicate that actively targeted MB can detect angiogenesis, inflammation, and thrombus formation [3]. In this context, even small changes in marker expression can be quantified by molecular US. Undoubtfully, molecular US imaging has high potential for clinical translation, and two targeted MB formulations are already in clinical trials. Complementary to MB, there is increasing research activity on molecular US imaging using nanobubbles (NB) and their targeting to extravascular markers [4]. UCA are also employed for US mediated drug and gene delivery. Drug or gene uptake is facilitiated by increasing the permeability of cell membranes and biological barriers with oscillating or bursting MB. However, this topic is already addressed in recent review articles [5,6,7].

Therefore, this review article focus on the evolution of molecular US, highlighting the newest molecular US imaging technologies, providing an overview of targeted contrast agent formulations, and summarizing their applications in preclinical and clinical studies. In the context of the latter, this paper also discusses challenges that have to be overcome to accelerate clinical translation. 

## 2. Detection Technologies

UCA can be injected into the blood pool to increase the contrast during imaging. The detection of UCA relies on their non-linear response to US. When the sound waves are emitted from the transducer and hit the UCA they are backscattered with a different frequency (non-linearity), while for tissues the emitted and returning signals are more linear. By this, the signal from the UCA can be distinguished from the tissue signal (Figure 1).

This nonlinearity originates from the nature of the response of the UCA to the US wave, which consists of high- and low-pressure phases. When the UCA is exposed to high pressure, it is compressed, and when it is exposed to low pressure it expands. With high wave amplitudes, the compression during the rise in pressure is bigger than the expansion during the pressure drop, which creates the non-linear response.

There are several techniques using non-linearity to detect UCA such as pulse/phase inversion, power modulation, or contrast pulse sequencing (Figure 2).

During pulse/phase inversion two pulses are transmitted. The second pulse is shifted by 180 degrees. Afterward, the responses to these pulses are summed up. For mostly linear responses as it is the case for tissue, the total is close to zero. For UCA the response does not cancel out due to irregularity in the backscattered US wave (non-linearity). 

During power modulation, two identical pulses are sent out with a two-fold difference in amplitude. The response pulses are then summed up, while the second pulse is multiplied by two. With a linear response from tissue the total is again close to zero. With UCA, the total is not zero due to a difference in shape and amplitude (non-linearity) [8].

Contrast pulse sequencing is a combination of both above mentioned methods. Practically, during contrast pulse sequencing two pulses are used, where the second one is shifted 180 degrees and has an amplitude twice the magnitude as the first one. The advantage of this technique is that it can be used at low pressure [9,10].

While the imaging techniques mentioned above do not destroy the UCA, there are methods, which destroy the MB by US. In the field of molecular US imaging, a prominent example is the 3D SPAQ method.

SPAQ can be applied in cases where high local densities of bound MB are present in the tissue but the blood pool is already cleared from free circulating MB. After injection, the targeted MB are allowed to bind to their target and unbound MB are allowed to wash out (approximately after 10 min). Then, a destructive US pulse is applied in the tissue of interest. When MB are destroyed the resulting non-linear signal is construed as a strong movement and detected by Doppler imaging. Subsequently, the transducer is moved forward in the micrometer steps and Doppler imaging is applied to destroy the MB. From the second destructive pulse, MB are only present in the non-overlapping part of the sound field, which should be much smaller than the voxel size. Thus, signals of multiple MB within one voxel can be assessed separately. At the end, a 3D data set is generated that displays MB destruction events with higher resolution than is possible with a single image assessment [12].

Alternatively, to detect targeted MB the destruction - replenishment technique can be used. Targeted MB are injected and images are recorded. After the targeted MB are expected to have bound to their target, a high mechanical index pulse is applied that destroys the MB in the examined area. Images are recorded promptly after the destructive pulse. By subtracting the mean signal intensity of images after the destructive pulse from that directly before, the signals originating from target-bound MB can be assessed [6,13].

Another approach was suggested by Pysz and colleagues where targeted MB are identified by their dwelling time in one spot. To do so, the pixel intensities representing the MB in each frame during B-mode imaging are monitored over a specific time frame. First, it is evaluated whether there is an increase in pixel intensity indicating the presence of a MB. Then, it is evaluated if the pixel intensity stays constant and a time threshold is defined for which a MB is considered as target bound. This technique can be applied in real-time with minimum post-processing and without using a destructive pulse exposing the tissue to a high acoustic pressure [14].

## 3. Targeted Contrast Agents

The main UCA are MB, NB, and nanodroplets (Figure 3). To enhance their specificity, various ligands were coupled, including antibodies, peptides, and carbohydrates. Depending on their size, UCA have been targeted to intravascular or extravascular markers. 

### 3.1. Functionalization of Contrast Agents

The easiest way to functionalize UCA is to have the targeting ligand incorporated in the shell layer of the bubble (Figure 4). This method favorably works for lipid-bound ligands such as phosphatidylserine [15] and phospholipid-heteropeptides binding to the vascular endothelial growth factor receptors 2 (VEGFR2) [16], because they can withstand harsh MB synthesis conditions.

Another way of functionalizing UCA is to incorporate a reactive moiety into the shell that can be coupled via a non-covalent or covalent bond to the targeting ligand. For non-covalent coupling, an avidin-biotin bond has been extensively used. In this case, biotin is incorporated into the shell layer and with the use of avidin as a “bridging” moiety biotinylated ligands are coupled. Since avidin has four possible binding sites multiple ligands can be attached. Additionally, it was shown that avidin itself can be incorporated into the bubble shell layer. The major drawback of using a biotin-avidin bond for UCA functionalization is the immunogenicity of avidin. Thus, avidin-biotin bubbles can only be used in preclinical research and not in the clinics.

Furthermore, two covalent binding methods (Figure 4) for UCA functionalization have been used. The first one is the carbodiimide coupling. In this case, the carboxyl group located on the shell coat reacts with an amine group present in the targeting ligand forming an amide bond. Due to the low yield of the reaction, an excess of ligand has to be used, which can become expensive [17]. Additionally, if the ligand contains multiple amine groups (as it is the case for most proteins) the coupling can occur at several sites leading to uncontrolled conjugation and a possible reduction in targeting affinity. Hence, the second method using maleimide-thiol coupling is preferred. Maleimide can be easily conjugated to polyethylene glycol (PEG)-lipids that can be used for the bubble synthesis, while the thiol group has to be attached to the ligand. In single-step “click” reaction maleimide and thiol groups form a thioether bond. This method has higher yield compared to carbodiimide coupling, thus fewer ligands are needed. Moreover, due to the single thiol group introduction to the targeting ligand the coupling to the UCA is better controlled. Though covalent bond avoids immunogenic materials like avidin, the unreacted chemical groups on the shell layer might as well trigger an immune response. Thus, it is important to have all chemical groups bonded.

In some cases, PEG was used as a spacer between the bubble and the targeting ligand (Figure 4). A long chain of PEG gives the targeting ligand more flexibility and mobility to interact with its receptors. It has been shown that ligand interaction to its receptor was increased with the increasing length of the spacer [18,19]. Moreover, the ligand could be shielded by introducing additional longer PEG chains without targeting moieties [20,21]. This prevented specific interactions with the target. Only when US irradiation was applied the longer PEG chains unfolded and exposed the ligand to its receptor. Then MB were able to bind to the target. Thus, the bimodal structure produces stimulus-responsive, targeted UCA. The advantage is reduced immunogenicity and low binding at not insonated sites.

### 3.2. Intravascular Targeting (MB)

Due to their size, MB do not leave the vasculature, so targets should be located within the vessels. This is the case for many angiogenesis, inflammation, and thrombosis markers, for which several ligands have been investigated (Table 1). 

Due to the fast flow of MB in larger vessels, it is important that the ligands bind quickly to the target. For this purpose, a high kinetic association for binding is needed. Alternatively, an increase in binding can be achieved by using multiple ligands on one bubble (more about it in Section 3.2.4) or by applying acoustic radiation force pulses.

Acoustic radiation forces (primary and secondary) are the forces affecting MB localization and distribution. They cause movement of MB (primary radiation force) and interaction of the MB with each other (secondary radiation force) [98,99]. The primary radiation force is experienced by single particles resulting from the acoustic pressure field. It leads to the movement of the MB in the acoustic field and also allows to push a MB streamline towards the vessel wall bringing the bubbles closer to the target, which might increment targeting efficiency [99,100,101]. In this regard, Dayton et al. showed that during this process the flow of the MB concentrating close to the vessel wall was reduced compared to the MB floating in the streamline [100]. Secondary or Bjerknes force affects neighboring bubbles. It is produced by the scattered field of a resonating bubble. This leads to a reversible attraction and clustering of MB [99,100]. Moreover, the clustering of MB is affected by the distance between two bubbles [98]. It is hypothesized that bound bubbles can attract other bubbles thus increasing targeting efficacy and the concentration of bound MB [100,101]. Following up, Zhao and colleagues showed in vitro that enhanced binding of targeted MB was observed when radiation force was applied [101]. In line with this, the in vivo study from Gessner et al. using a cyclic arginine–glycine–aspartic (RGD) peptide targeted to α_v_β_3_ demonstrated a significant increase in targeting efficacy and US signal intensity, when acoustic radiation force was applied [102]. In addition, Wang et al. programmed a conventional US imaging device with an acoustic radiation force inducing sequence for pushing the MB to the vessel wall by using low pressure and long duty cycles. The experiments successfully demonstrated higher MB binding to P-selectin in large blood vessels in vitro and in vivo, where good binding efficiency is usually difficult to achieve due to low contact between MB and the vessel wall at the high physiological flow rates [103]. Furthermore, with the introduced sequence the signal between molecularly adherent, non-specific adherent and free floating MB can be distinguished [104]. This method could be very helpful since all molecular US imaging methods reported above cannot distinguish between unspecific adherent stationary MB and target-bound ones, which increases the unspecific background signal.

#### 3.2.1. Angiogenesis

Angiogenesis describes the process of vessel formation and is crucial for tumor growth and metastasis. Among angiogenesis markers, VEGFR2 and α_v_β_3_ integrin are the most addressed targets. VEGFR2 also known as KDR is an endothelium-specific receptor, that is highly expressed on tumor-associated endothelial cells [105]. Activation of the VEGFR2 pathway triggers multiple signaling cascades that result in endothelial cell survival, mitogenesis, migration, differentiation, and alterations in vascular permeability [103]. Moreover, overexpression of VEGFR2 has been linked to tumor progression and poor prognosis in several tumors [106]. Integrins are transmembrane receptors that are expressed on endothelial and tumor cells. They activate signaling cascades that regulate gene expression, cytoskeletal organization, cell adhesion and cell survival. This facilitates tumor growth, invasion and metastasis [107]. In particular, α_v_β_3_ integrin expression is low on endothelial cells under normal conditions, but is elevated during tumor angiogenesis [108,109,110]. Another angiogenesis related marker is Endoglin, a transmembrane glycoprotein expressed in proliferating endothelial cells such as tumor endothelial cells [111]. It is a component of the transforming growth factor beta receptor complex involved in cell proliferation, differentiation and migration [111].

MB functionalized with ligands against VEGFR2, integrin and other angiogenesis markers have been successfully tested for molecular US imaging (Table 1). Moreover, the CEUS signal of the targeted MB could be correlated to the level of angiogenesis marker expression in the tissue. For example, CEUS signal using VEGFR2-targeted MB increased from hyperplasia to ductal carcinoma in situ and invasive breast cancer compared to normal tissue [46]. This was additionally confirmed by histological analysis.

During tumor angiogenesis, multiple endothelial markers are overexpressed [112]. These markers could be good targets for cancer detection. In this respect, E-selectin [86,87], secreted frizzled related protein 2 (SFRP2) [113], B7–H3 [54], nucleolin [55] and thymocyte differentiation antigen 1 (Thy1) [56] were tested for cancerous tissue detection and showed promising results. Since angiogenesis is an early event in tumor development [114], angiogenesis targeted MB should be able to detect tumors at early stages. Indeed, ovarian cancer at an early stage was sensitively detected using α_v_β_3_ integrin-targeted MB [27]. Also, VEGFR2-targeted UCA were able to visualize breast cancer tumors as small as 2 mm in diameter [44,46] and pancreatic ductal adenocarcinoma lesions smaller than 3 mm in diameter [47]. Interestingly, a study showed that smaller breast cancer xenografts express the highest amount of VEGFR2, while with increasing tumor size the expression decreases [44]. Thus, current results suggest that VEGFR2 targeted MB are promising UCA for early tumor detection, though more research should be done.

Since angiogenesis plays an important role in tumor growth, it is not surprising that multiple anti-cancer drugs target angiogenesis. The idea is to stop tumors from developing new blood vessels and hopefully shrink them by cutting the nutrient and oxygen supply. CEUS imaging using targeted MB can provide information on the angiogenesis profile of the tumor and assess antiangiogenic therapy effects. Several groups showed successful monitoring of antiangiogenic therapy using VEGFR2-targeted MB [29,31,35,45,49,50]. In all cases, a clear decrease of VEGFR2-targeted MB accumulation was observed after treatment compared to untreated controls. These findings were further confirmed with immunohistochemistry analysis. Moreover, other therapies such as gemcitabine [35], nilotinib [45], and carbon ion treatment [28] were monitored using CEUS with angiogenesis targeted MB. Similarly, as for antiangiogenic therapy monitoring, CEUS was correlated to the target marker expression according to immunohistochemistry analysis. Hence, CEUS using angiogenesis targeted MB seems to be a promising tool for non-invasive antitumor therapy monitoring.

Furthermore, by combining functional and molecular US imaging, vascular responses in tumors can be comprehensively characterized. This is important since the percentage of angiogenic vessels – indicating angiogenic activity – cannot be assessed solely from the information on VEGFR2 bound MB. For, example, Palmowski and co-workers showed that VEGFR2 and α_v_β_3_ integrin-targeted MB bound less after administration of a matrix metalloproteinase (MMP) inhibitor, which was in line with immunohistochemical analysis. However, when normalizing the molecular marker expression to vascular density no change in the percentage of angiogenic vessels was visible indicating that the MMP inhibitor did not decrease angiogenesis but induced a general decrease in vascularization [29]. To unravel these effects by US, Bzyl et al. injected long-circulating UCA (BR38) to derive functional information on vascularization and relative blood volume and VEGFR2-targeted MB (BR55) to assess angiogenesis [39]. Two breast cancer models with different aggressiveness and angiogenic activity were evaluated. The results showed that the more aggressive tumor model (MDA-MB-231) had the higher total expression of VEGFR2 compared to MCF-7 according to VEGFR2-targeted US measurements and immunohistochemistry. Also, the normalization of the molecular imaging data to the relative blood volume confirmed the higher angiogenic activity of the MDA-MB-231 model. A few years later Baetke et al. showed that functional and molecular US imaging can even be performed in one examination using only the targeted MB [49]. Here, antiangiogenic therapy effects were monitored in squamous cell carcinomas using BR55. It was shown, that the first-pass analysis of VEGFR2-targeted MB was not strongly affected by the targeting, and vascularization results were comparable to those obtained with non-targeted MB. Hence, in the early binding phase, functional information could be obtained and at the late phase, the molecular angiogenesis profile of the tumors could be assessed (Figure 5). Combining functional and molecular US in one examination would reduce measurement time and the need for multiple injections and multiple formulations of MB.

In summary, all these pre-clinical studies using angiogenesis-targeted MB showed promising results confirming that the targeted UCA can be used for angiogenesis profiling in various tumors. Most research has been done using VEGFR2-targeted MB, especially using the clinical-grade contrast agent BR55. These MB can detect tumors at early stages and can be used for antiangiogenic therapy monitoring combining functional and molecular US. BR55 is now evaluated in clinical trials and the initial results are summarized in Section 3.2.5.

#### 3.2.2. Inflammation and Atherosclerosis

Atherosclerosis is an inflammatory disease that is driven by endothelial dysfunction. Many patients do not have symptoms for a long time until a plaque ruptures or severe narrowing or total blocking of the blood vessel occurs. Although there are biomarkers to identify atherosclerosis [115,116], tools that can predict the growth dynamics of a plaque or its risk to rupture are hardly available. CEUS with targeted MB could suit this purpose considering its high sensitivity and capability to quantify even low amounts of the inflammation-related markers. Vascular cell adhesion molecule 1 (VCAM-1) is known to be overexpressed and translocated to the luminal surface of endothelial cells at the early stages of atherosclerosis [117,118]. Thus, VCAM-1-targeted MB have been proposed for atherosclerosis imaging [63,64,65,66,67,68,69,70,71,73,74]. These UCA have successfully identified atherosclerosis and the CEUS signals could be correlated to the progression of the disease [65,67]. Moreover, molecular changes during apocynin [63,64,69] and statin [66] therapy were successfully monitored. Further, VCAM-1, platelet glycoprotein (GP) Ibα [63,64,69], junctional adhesion molecule A (JAM-A) [62], GP VI [94], lectin-type oxidized low-density lipoprotein receptor 1 (LOX-1) [74], and von Willebrand factor (VWF) [69,74] have been used for atherosclerosis imaging. Of these markers, JAM-A expression might be particularly associated with early plaque formation and vulnerability [119]. Indeed, in mice with partial ligation and atherogenic diet molecular US of JAM-A indicated atherosclerosis at a very early stage and was even capable of assessing vascular sites that underwent an immediate change in shear stress [61]. Moreover, another group demonstrated that plaque vulnerability can be assessed in rabbits [62]. Another interesting target is GP VI since it can be used for atherosclerosis therapy. It has been shown that administering soluble anti-GP VI antibody inhibits thrombus formation and progression of atherosclerosis [120,121,122]. Using high-frequency US, it is possible to disrupt MB and release the ligand coupled to the shell. Thus, GP VI-targeted MB could work as theranostic agents. A study showed that these MB help to diagnose atherosclerosis, and in combination with the high-frequency US work as a therapy [94]. 

Another pathology that would benefit from a broadly available reliable non-invasive detection technique is acute cardiac ischemia. Electrocardiogram and serological markers often have limited expressiveness or lead to misjudgment [123,124,125,126]. The gold standard in the clinics is cardiac magnetic resonance imaging (MRI) though it is time-consuming, requires multiple breath holds, and due to usage of gadolinium cannot be used in patients with renal dysfunction. Other diagnosis methods include scintigraphy and coronary CT. However, both techniques raise safety concerns due to usage of radioisotopes and/or radiation. Myocardial ischemia is associated with endothelial upregulation of adhesion molecules which persist after ischemia has resolved. Thus, it could be possible to identify post-ischemic myocardium using the molecular US. P-selectin seems to be a good target since it is expressed within minutes after ischemia or injury [127]. As expected, P-selectin-targeted MB strongly bound in post-ischemic myocardium in mice [74,83]. Similar results were also seen for E-selectin-targeted UCA in rats (Figure 6) [88]. Moreover, the researchers decided to target both, P- and E-selectins [84,85,128,129]. This proved to be particularly useful since P-selectin was present immediately after ischemia but only E-selectin was still overexpressed after 24 h [84]. Thus, the dual-targeted MB could detect myocardial ischemia for a longer time. Further in vivo experiments were performed in primates demonstrating that molecular US using selectin-targeted MB is a safe and effective method to detect myocardial ischemia [129]. However, despite these promising results, molecular US imaging of the heart is difficult to perform, user-dependent, and requires an experienced physician. Thus, it must be carefully evaluated whether a high diagnostic accuracy and reproducibility can also be achieved outside specialized centers and whether it can compete with the clinically established imaging methods. 

Inflammatory bowel disease (IBD) is currently assessed by endoscopic monitoring, US, and clinical chemistry. Endoscopy is invasive and cannot access major parts of the small intestine. US is applied to capture the enhanced thickness of the inflamed bowel wall and the enhanced perfusion by Doppler. However, this approach has limited sensitivity and only becomes prominent at advanced disease stages. Here, molecular US could be introduced as a complementary sensitive tool. Inflammation in patients with IBD is associated with increased expression of cell adhesion molecules. Initially, mucosal addressin cell adhesion molecule 1 (MAdCAM-1) targeted MB were proposed for IBD detection [60]. The CEUS signal using these MB positively correlated with the severity of ileal inflammation assessed by immunohistochemical analysis. However, no further research was published using these MB. Instead, molecular US of IBD was mostly approached using different selectin-targeted MB. For example, P-selectin-targeted MB were successfully applied for IBD detection and monitoring of anti-TNFα antibody therapy [81]. CEUS signal changes were observed already after 3 days of therapy, while there was no decrease in bowel wall thickness or perfusion yet. In another study, these MB were able to depict radiation-induced P-selectin expression in the colon [82]. Furthermore, clinical-scale P- and E-selectin-targeted UCA were tested for IBD detection. The experiments in colitis induced mice showed that molecular US assessment of IBD excellently correlated with ^18^FDG-PET (positron emission tomography) and histological examination [130]. Further studies in swine demonstrated that dual-targeted UCA have the potential for clinical translation [131,132]. Wang and co-workers showed that, one hour after inflammation induction by exposing ileum to 2, 4, 6-trinitrobenzene sulfonic acid (TNBS) and ethanol, a significant increase in CEUS signal occurred [131]. The US signal changed in line with the increased selectin expression and further increased with the progression of the inflammation. Moreover, P- and E-selectin-targeted MB also proved promising for long-term monitoring of anti-inflammatory IBD treatment in swine [132]. The combination treatment of prednisone and meloxicam reduced inflammation and downregulated selectin expression in the inflamed vasculature in the bowel. This molecular change was clearly depicted by molecular US imaging. Interestingly, there was no difference in the inflammation score obtained from the histological analysis between treated and control groups. This underlines the high sensitivity of selectin-targeted US imaging for IBD assessment and highlights its potential for clinical translation.

#### 3.2.3. Thrombosis

Thrombosis describes the formation of a blood clot in a vessel. In the clinics, Doppler US is a standard method for detecting deep venous thrombosis (DVT). However, the blood clot has to be big enough to produce visible circulation defects and the thrombus activity cannot be assessed. Moreover, the examination is user-dependent and hence the accuracy in detecting DVT varies [133]. MB targeting activated platelets have been proposed for thrombus detection and characterization. Acute thrombus should have a higher number of activated platelets than the chronic state. Hence, ligands, which only bind to activated platelets, have been proposed for targeting: lysine-glutamine-alanine-glycine-aspartate-valine (KQAGDV) [89,90], glycoprotein (GP) IIa/IIIb [93] and cyclic arginine-glycine-aspartate (RGD) [91,92]. All targeted MB formulations showed thrombus-specific US contrast in vitro and in vivo. Nonetheless, the exact concentration of activated platelets required to successfully identify a thrombus by targeted MB still needs to be elucidated, which is crucial for evaluating the potential of the method for early thrombus detection.

Furthermore, stimulus-responsive MB have been suggested for thrombus detection. The first group proposed MB decorated with aptamers [95,96]. When exposed to thrombin, the aptamers were detached and the bubble stiffness was reduced. The change in shell stiffness alters the harmonic signals generated by the bubble. For thrombus formation a critical concentration of 25 nmol is required [134,135]. At this concentration it was seen that all aptamers detach from the MB making the shell soft, while at lower concentrations (10 to 25 nmol) only a partial softness change was observed. Hence, these MB become active and give US contrast only when thrombin levels exceed the critical threshold. In vitro and in vivo experiments proved the concept. The stimulus-responsive MB increased the CEUS signal five-fold in presence of clots and even small lesions that were not visible by non-specific UCA were detected. Another group proposed MB with thrombin-sensitive activatable cell-penetrating peptides (ACPP) [97]. In this case, thrombin cleaved ACPP making the UCA positively charged. Due to Coulomb interaction, activated MB adhere to negatively charged surfaces like red blood cells, fibrin, and platelets. The concept of these UCA was confirmed in vitro using rabbit blood and an increase in the US signal was only seen if thrombin was present (Figure 7). Therefore, ACPP MB detect only acute blood clots. Both stimulus-responsive MB formulations can distinguish between acute and chronic thrombosis according to thrombin levels in the blood.

#### 3.2.4. Multiple Targets

Besides single targeted UCA, few groups have synthesized and tested multi-targeted ones (Table 2). The idea is that addressing multiple targets at the same time improves MB binding affinity and avidity. This would reduce the amount of unbound bubbles, meaning a lower concentration would be needed to achieve similar results as with single-targeted MB, and the bubbles would persist for a longer time. Furthermore, in the case of a disease with heterogeneous expression of the individual markers, multi-targeted UCA could improve the accuracy of diagnosis.

First multi-targeted MB were functionalized using anti-ICAM-1 antibodies and sialyl Lewis X [133]. These dual-targeted MB had greater adhesion to inflammatory endothelial cells compared to their single targeted counterparts under shear flow. Similar results were reported for MB modified with anti-VCAM-1 and anti-P-selectin antibodies in flow chamber experiments [137]. Moreover, VEGFR2 and α_v_β_3_ integrin-targeted MB for angiogenesis imaging showed the benefits of dual-targeting in vivo [138]. Since dual-targeted MB comprise of two ligands it was expected that its binding affinity and avidity would be an average of the single-targeted MB counterparts. Interestingly, in all three studies, dual-targeted MB showed significantly higher binding affinity than their single targeted counterparts. Hence, the ligands on MB seem to synergistically increase the binding affinity.

The main drawback of dual-targeted UCA is the synthesis route. In the case of attaching two ligands to the MB, a ratio between the ligands has to be chosen. Controlling this ratio during synthesis can be difficult. This challenge can be overcome by using a single ligand that targets multiple markers, such as sialyl Lewis X that binds to P- and E-selectin. MB decorated with this natural ligand identified postischemic myocardium better than UCA targeted to just P- or E-selectin [128]. Moreover, even better results were achieved using P-selectin glycoprotein ligand-1 analog (PSGL-Ig) instead of sialyl Lewis X [80]. PSGL-Ig-targeted MB enabled the detection and quantification of inflammation in colitis induced mice [130] and swine [131,132]. Moreover, post-ischemic myocardium was detected in mice [84], rats [85], and even macaques [129]. Hence, inflammation detection using P- and E-selectin-targeted MB is possible and preferential over single-targeted UCA. 

Furthermore, MB modified with VEGFR2, α_v_β_3_ integrin and P-selectin have been synthesized [139,140]. These triple-targeted UCA possess better binding to target cells than single- or even double-targeted MB (Figure 8). The in vivo experiments further confirmed these results. Interestingly, in vivo US intensity using triple-targeted UCA was higher than the cumulative intensity of all single-targeted MB [139]. Similar to dual-targeted MB, the ligands synergistically increased the efficiency of the binding. Moreover, these MB were used for monitoring antiangiogenic therapy in breast cancer-bearing mice [140]. Due to the high US signal produced by multi-targeted UCA, even small changes in angiogenesis could be detected. Therefore, an early response to the therapy can be assessed. However, no further studies using this triple-targeted MB have been reported so far.

#### 3.2.5. Targeted MB in The Clinics

Although there are many successful pre-clinical studies using targeted MB, the translation to the clinics is slow. Most of the targeted MB used in research have been functionalized using biotin-avidin or covalent coupling. As discussed before, avidin can cause severe allergic reactions in patients and thus its use should be avoided [141,142,143,144,145]. However, free chemical groups sticking out of the MB shell can as well cause an immune response. Thus, it is not surprising that the only two targeted MB (Sonazoid™ and BR55) currently in clinical trials have the ligand incorporated in the shell layer.

Sonazoid™ (Daiichi Sankyo Company Ltd.) is not specifically advertised as a targeted MB. However, due to phosphatidylserine in the shell layer, these MB circulate for a longer time in the blood and accumulate in Kupffer cells and macrophages of the liver. These MB are clinically approved in Japan for CEUS imaging of hepatic tumors. They are also in phase I clinical trial in the United States (ClinicalTrials.gov identifier: NCT02968680) for detecting sentinel lymph nodes in patients with melanoma and in phase III clinical trial in China (NCT03335566) for liver lesion detection. 

The second targeted MB in the clinical translation is BR55 (Bracco Suisse SA, Geneva, Switzerland). These MB contain VEGFR2-binding phospholipid-heteropeptides. The first clinical trials in women with ovarian or breast cancer showed that BR55 is safe to use in patients, and the CEUS signal correlated with the VEGFR2 amount in tumor lesions [146]. Further clinical trials are ongoing to evaluate its specificity and sensitivity in the prostate (NCT02142608 and NCT01253213), ovarian cancer (NCT03493464 and NCT04248153) and pancreatic lesions (NCT03486327).

### 3.3. Extravascular Targeting (NB)

Since MB are trapped in the bloodstream due to their size, extravascular imaging is not possible. This is not necessarily a drawback, since many promising targets are still accessible. Furthermore, the intravascular distribution minimizes the unspecific MB accumulation in tissues leading to a low background. However, if extravascular targeting is desired one can adapt the UCA size to the size of the vascular fenestrations. In leaky tumor vessels fenestrations are known to be around 400–800 nm large [147]. Therefore, bubbles in the nanoscale were introduced, which can leak from the vasculature into the tumor tissue [148,149,150]. Due to their accumulation via the enhanced permeability and retention effect (EPR), they show prolonged persistence at tumor site compared to MB. By targeting over-expressed tumor markers, the retention of the NB in the tumor tissue can be enhanced [7,151]. An overview of targeted NB is provided in Table 3.

Due to their small size, NB are expected to backscatter less and, thus, show less signal during US imaging [159]. Furthermore, for smaller bubbles higher frequencies are needed, which can hamper imaging, since high frequencies do not reach deep into tissue [160]. Moreover, US devices used in the clinics usually do not have the transducers for high frequencies.

Surprisingly, some studies report that NB show similar echogenicity as MB (Figure 9) [148,161]. There are no clear explanations for this observation. However, one group discusses that there might be an underestimation of NB concentrations in the phantoms because NB are difficult to count [161]. 

Furthermore, highly flexible shells of some NB may increase the echogenicity significantly, even at lower frequencies than their resonance frequency [159,160,161,162]. For instance, Perera and colleagues designed a new shell, which was inspired by nature, where shells with several layers exist. The layers differ in their elastic properties such as in bacterial cell envelopes [163]. Perera adapted this concept and designed a shell consisting of two layers differing in elasticity, one compliant layer, another stiff adlayer with PEG on top. This composition provides high shell stability, which reduces air loss during oscillation and improves better persistence in the blood circulation. Together, these properties promise an extended effective visualization of the tumor.

By targeting NB a longer tumor persistence but not necessarily a big difference in US signals was observed [152,157]. This is in line with results on other nanoparticles, were targeting only improved retention but not accumulation, the latter being mostly mediated via the EPR effect [164]. Thus, obtaining exact information about extravascular molecular marker expression with particles of this size will remain difficult. However, the exploration of the diagnostic benefits resulting from NB accumulation is just beginning and needs further research and reasonable conceptual considerations.

### 3.4. Phase Shift Nanodroplets

Nanodroplets were introduced as UCA to prolong circulation time and enhance extravasation to tissue. Since their application mainly focuses on drug delivery rather than imaging, only a brief overview of their functional principle is given in this review.

Nanodroplets have a perfluorocarbon liquid core stabilized by lipids, polymers, or proteins [165]. The emulsion is injected in liquid form, not showing much contrast during US imaging compared to MB. However, when US is applied as an external trigger, the core solution shifts from liquid to gas phase forming MB and achieving high echogenicity [166,167]. This process is called acoustic droplet vaporization (ADV). This comes with several advantages. The solution in its liquid phase has a higher circulation time and can leave the vasculature. Additionally, the phase shift can be activated at the target site. Two elements are important for ADV, namely the temperature to reach the boiling point of the liquid and the pressure. The boiling point is usually already reached at body temperature. For instance, the boiling point for perfluorocarbon gas starts from 29 °C. Hence, small bubbles might already be forming when the emulsion is injected in the body. By additionally applying the US, the surrounding pressure is lowered and less than the vapor pressure, this enhances the phase shift and formation of MB [165,168]. 

A chemotherapeutic drug can be added to the emulsion and thus, encapsulated in the MB after ADV. By using US, the MB collapse and the drug will be released at the target site. Since the nanodroplets are capable of extravasating into tumor tissue, they can be used for contrast-enhanced tumor imaging combined with drug delivery all in once [169,170,171]. Furthermore, there have been targeted nanodroplets approaches [172,173,174]. As targeting ligands, folate [172,173] and anti-Her2/*neu* peptide [174] were used. By adding these targeting ligands, the formed bubbles stayed attached at the vaporized area visualizing the subject area aimed for therapy. However, more research is needed to evaluate the added value of targeted nanodroplets over NB or MB regarding their imaging abilities. 

## 4. Conclusions

Molecular US has broadened the application spectrum of standard US. Multiple effective endovascular targeting strategies have been explored using MB, and are now complemented by NB that open access to extravascular targets. Unfortunately, of the many pre-clinically explored actively targeted MB, only few meet the criteria for clinical translation. Furthermore, important questions about MB pharmacokinetics and the fate of shell components, but also about upscaling of MB production, batch reproducibility, storage stability, and sterilization, often remain open. Without these questions being answered by our scientific community, it will be difficult to convince investors or pharmaceutical industry to consider these agents as potential clinical products. In addition, the clinical indications must be very carefully chosen. High comfort to the patient, lack of radiation, excellent availability of US devices, and low costs are arguments speaking for the clinical implementation of molecular US imaging. However, to have a chance for broad acceptance, the molecular US methods need to be at least equivalent to or able to provide a clear added value over the clinical diagnostic standard methods. This can be reflected in higher sensitivity and specificity in disease detection, classification, and therapy response monitoring. Taking these arguments into consideration, indications in the fields of angiogenesis in cancer, inflammatory diseases, and arteriosclerosis/thrombosis appear highly meaningful. Ideally, the chosen applications should already use US in the clinical diagnostic routine. The molecular US examination then increases the power and expressiveness of the routine examination, enables faster therapeutic decision making, and reduces the need for further diagnostic analyses. Unfortunately, often, the most promising indications do not address huge markets, and thus are financially not interesting for big pharmaceutical companies. In addition, academia faces the dilemma that third-party funding is often only available for basic research and clinical phase II/III studies, but hard to get for closing the critical translational gap in between. Nonetheless, to move this exciting technology into clinical application, in our opinion, the translational process must be actively promoted by academia together with spin offs and small and medium enterprises, for which smaller markets are still attractive. Then, molecular US imaging could evolve to its full potential as a highly sensitive, specific, and real-time molecular imaging tool to the benefit of patients suffering from various diseases.

## Figures and Tables

**Figure 1 nanomaterials-10-01935-f001:**
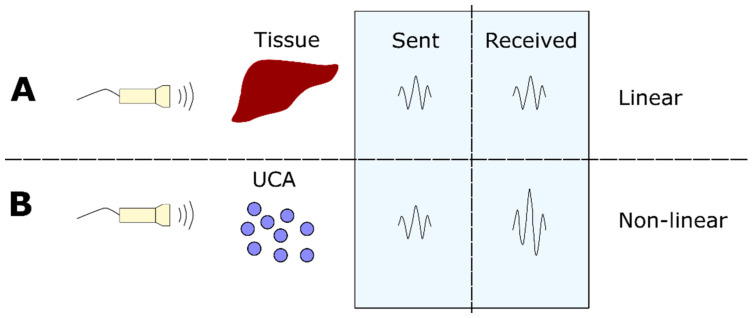
Non-linear response from UCA to US. The US transducer is transmitting ultrasound waves to tissue and UCA. For tissue (**A**) the sent and received signals are similar (linear response), while for UCA (**B**) the received signal has a different frequency than the initial frequency (non-linear response).

**Figure 2 nanomaterials-10-01935-f002:**
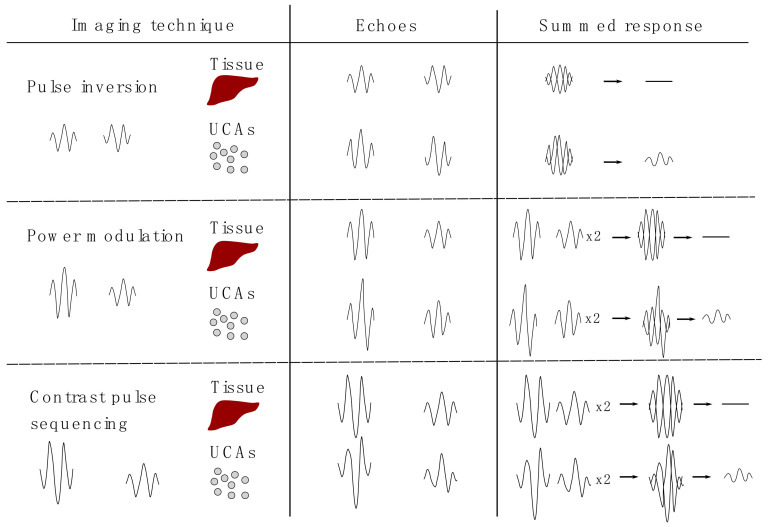
Theoretical background of imaging techniques based on the non-linear response of UCA. During pulse inversion two pulses are transmitted. The second pulse is shifted by 180 degrees. After receiving the echoes, the responses are summed up. In tissue the transmitted and received signals are the same and thus cancel out. For UCA the summed signal does not cancel out due to the non-linear response. During power modulation two pulses are transmitted. The second pulse has a two-fold difference in amplitude. For the summation of the responses the second pulse is multiplied by two. For tissue the response is near zero, for UCA the summed response is not zero due to the irregularity in UCA echoes. Contrast pulse sequencing is a combination of both mentioned methods. Two pulses are used where the second one is shifted by 180 degrees and has an amplitude twice the magnitude as the first one. Reproduced with permission from [11]. Copyright Elsevier, 2011.

**Figure 3 nanomaterials-10-01935-f003:**
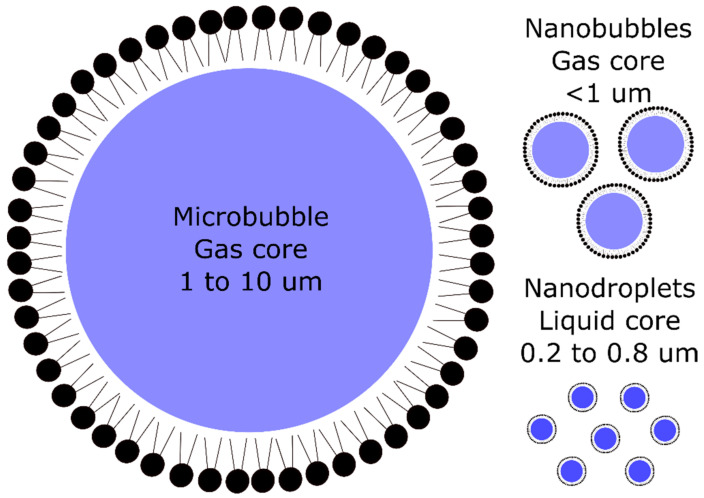
Schematic representation of microbubbles, nanobubbles, and nanodroplets.

**Figure 4 nanomaterials-10-01935-f004:**
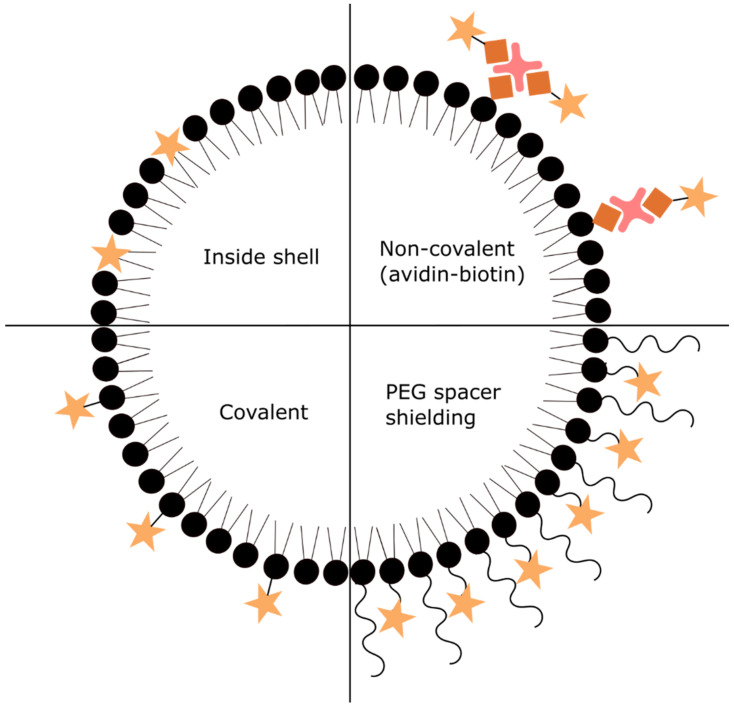
Schematic representation of different functionalization methods of UCA.

**Figure 5 nanomaterials-10-01935-f005:**
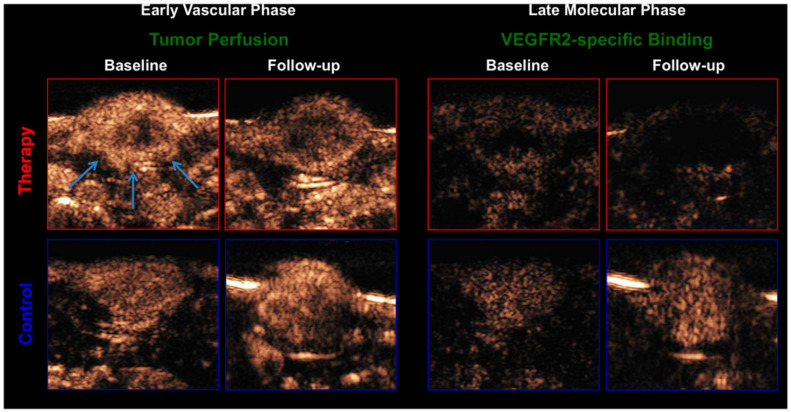
CEUS images of treated (top) and untreated tumors (bottom). Left side: early vascular phase with VEGFR2-targeted MB as a functional imaging biomarker; Right side: late phase of VEGFR2-specific binding with the targeted MB as a molecular imaging biomarker 8 min after contrast injection. Reproduced with permission from [51]. Copyright Eschbach et al., 2017.

**Figure 6 nanomaterials-10-01935-f006:**
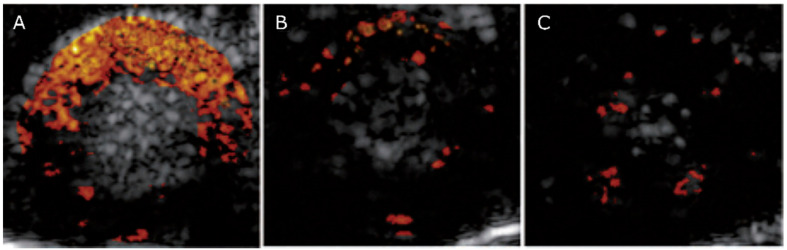
B-mode US images of the left ventricle 4 h after conorary occlusion overlayed with the signal of E-selectin targeted MB (**A**), non-targeted MB (**B**) and non-specific IgG targeted MB (**C**). Enhanced contrast signal is seen using E-selectin targeted MB. Reproduced with permission from [88]. Copyright SAGE Publications, 2014.

**Figure 7 nanomaterials-10-01935-f007:**
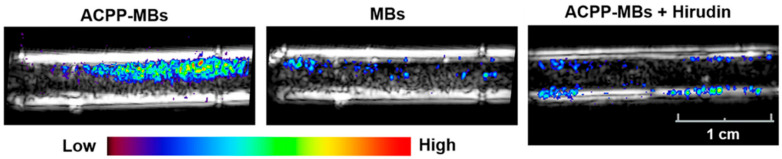
Molecular US imaging of thrombosis. Spatial maps of the molecular US signal color-coded and overlaid on B-mode images acquired at baseline after infusion of MB and washing with saline using targeted MB (left), non-targeted MB (middle), and targeted MB co-injected with hirudin (right). Targeted MB give specific signal only in the presence of thrombin. Reproduced with permission from [97]. Copyright American Chemical Society, 2017.

**Figure 8 nanomaterials-10-01935-f008:**
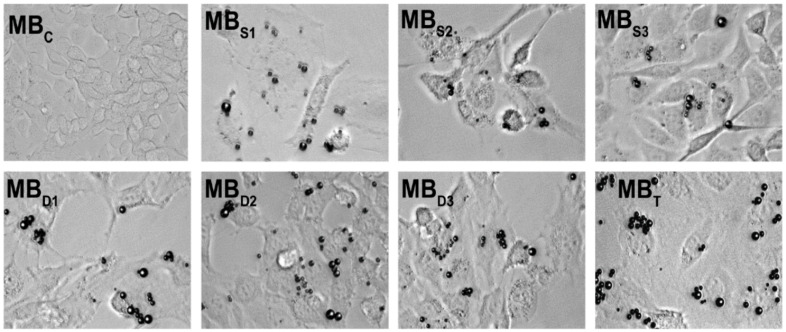
In vitro microscopy images of control (MB_C_), single (MB_S_), double (MB_D_), and triple (MB_T_) targeted MB adhered on cells. Triple targeted MB adhere more on the cells than single or double targeted ones. Reproduced with permission from [139]. Copyright John Wiley and Sons, 2011.

**Figure 9 nanomaterials-10-01935-f009:**
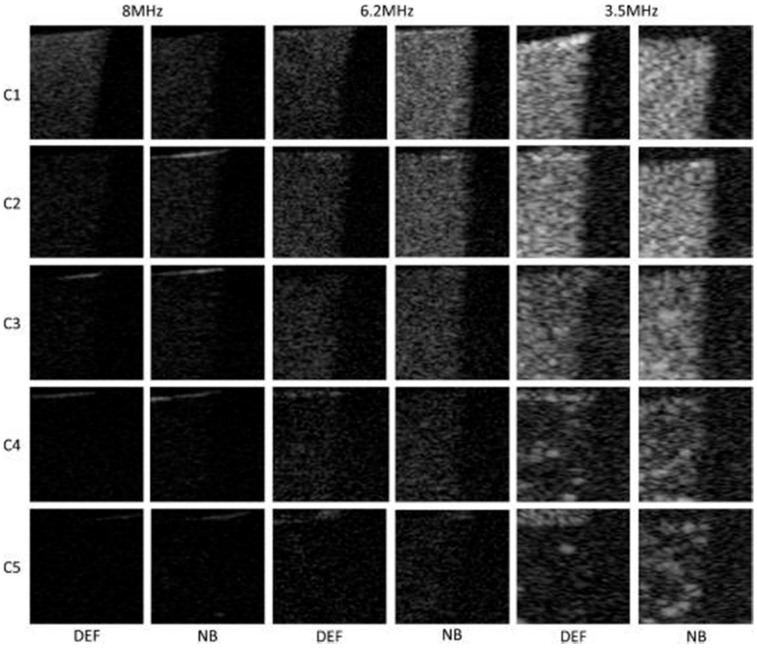
Contrast harmonic images of NB and commercially available Definity^®^ MB (DEF) at different concentrations and frequencies. Reproduced with permission from [161]. Copyright Elsevier, 2013.

**Table 1 nanomaterials-10-01935-t001:** Summary of intravascular targets investigated for molecular US imaging.

Target	Binding Ligand	Model
α_v_β_3_ integrin	Echistatin peptide	Malignant glioma in rats [22]
		Cremaster muscle in mice [23]
		Ischemic muscle in rats [24]
	Knottin peptide	Ovarian cancer in mice [25]
	Cyclic RGD peptide	Breast cancer in mice [26]
		Spontaneous model of ovarian cancer in hens [27]
	RGD peptide	Prostate cancer in rats [28]
		Squamous cell carcinoma in mice [29]
	Anti-α_v_β_3_ integrin antibody	Breast, ovarian and pancreatic cancers in mice [30]
		Skin cancer in mice [31]
		Cremaster muscle in mice [23]
	Cyclic RRL peptide	Prostate cancer in mice [32]
VEGFR2	Anti-VEGFR2 antibody	Squamous cell carcinoma in mice [29]
		Ovarian cancers in mice [30]
		Skin cancer in mice [31,33]
		Colon cancer in mice [34]
		Pancreatic cancer in mice [30,35]
		Malignant glioma in mice [36]
		Angiosarcoma in mice [36,37]
		Breast cancers in mice [30,38,39]
	10th type III domain of human fibronectin	Transgenic breast cancer in mice [40]
	Single-chain VEGF construct	Colon cancer in mice [41]
	VEGFR2-binding phospholipid-heteropeptides	Prostate cancer in rats [42]
		Breast cancer in rats [16] and mice [43,44,45]
		Transgenic breast cancer in mice [46]
		Transgenic pancreatic ductal cancer in mice [47]
		Colon cancer in mice [48]
		Squamous cell carcinoma in mice [49]
		Colon cancer in mice [50,51]
Neuropilin-1	CRPPR and ATWLPPR peptides	Pancreatic cancer in mice [35]
Endoglin	Anti-endoglin antibody	Breast cancer in mice [30,52]
		Ovarian cancers in mice [30]
		Pancreatic cancer in mice [30,35]
		Skin cancer in mice [31]
SFRP2	Anti-SFRP2 antibody	Angiosarcoma in mice [53]
B7-H3	Anti-B7-H3 antibody	Breast cancer in mice [54]
Nucleolin	F3 peptide	Breast cancer in mice [55]
Thy1	Anti-Thy1 antibody	Transgenic and implanted pancreatic cancer in mice [56]
Leukocytes	Phosphatidylserine	Inflammation in mice [57] and dogs [58]
	Anti-ICAM-1 antibody	Activated endothelial cells [59]
MAdCAM-1	Anti-MAdCAM-1 antibody	Inflammatory bowel disease in mice [60]
JAM-A	Anti-JAM-A antibody	Atherosclerosis in mice [61] and rabbits [62]
VCAM-1	Anti-VCAM-1 antibody	Atherosclerosis in mice [63,64,65,66,67,68,69,70] and swine [71]
	Nanobody targeting VCAM-1	Epidermoid carcinoma in mice [72]
		Atherosclerosis in mice [73]
	HGRANLRILARY peptide	Atherosclerosis in mice [74]
ICAM-1	Anti-ICAM-1 antibody	Endothelial cells [17]
		Inflammation in rats [75]
P-selectin	Anti-P-selectin antibody	Atherosclerosis in mice [63,65,69,70]
		Inflammation in mice [76,77] and flow chamber [78]
		Muscle inflammation in mice [79,80]
		Inflammatory bowel disease in mice [81,82]
		Myocardial ischemia in mice [83,84] and rats [85]
	LVSVLDLEPLDAAWL peptide	Atherosclerosis in mice [74]
	Sialyl Lewis X	Inflammation in mice [77]
E-selectin	IELLQAR peptide	Ovarian carcinoma in mice [86,87]
		Epidermoid carcinoma in mice [72,87]
	Anti-E-selectin antibody	Muscle inflammation in mice [80]
		Myocardial ischemia in rats [85]
	E-selectin affibody	Myocardial ischemia in rats [88]
GP Ibα	Anti-GP Ibα antibody	Atherosclerosis in mice [63,64]
	Dimeric murine recombinant A1 domain of VWF A1	Atherosclerosis in mice [69]
GP IIb/IIIa	Linear KQAGDV peptide	Thrombosis in flow chamber [89] and mongrels [90]
	Cyclic RGD	Thrombosis in mice [91,92]
	Anti-GP IIb/IIIa antibody	Thrombosis in mice artery [93]
GP VI	Anti-GP VI antibody	Atherosclerosis in mice [94]
VWF	Cell-derived peptide	Atherosclerosis in mice [69]
	RVVCEYVFGRGAVCS peptide	Atherosclerosis in mice [74]
LOX-1	LSIPPKA peptide	Atherosclerosis in mice [74]
Thrombin	Thrombin aptamer	Thrombosis in rabbit blood [95,96]
	Thrombin-sensitive ACPP	Thrombosis in rabbit blood [97]

**Table 2 nanomaterials-10-01935-t002:** Summary of multi-targeted MB investigated for molecular US imaging.

Targets	Binding Ligands	Model
ICAM-1 and selectins	Anti-ICAM-1 antibody and sialyl Lewis X	Flow chamber [136]
VCAM-1 and P-selectin	Anti-VCAM-1 and anti-P-selectin antibodies	Flow chamber [137]
VEGFR2 and α_v_β_3_ integrin	Anti-VEGFR2 and anti- α_v_β_3_ integrin antibodies	Ovarian cancer in mice [138]
Selectins	Sialyl Lewis X	Muscle inflammation in mice [80]
		Myocardial ischemia in rats [128]
	PSGL-Ig	Muscle inflammation in mice [80]
		Inflammatory bowel disease in mice [130] and swine [131,132]
		Myocardial ischemia in mice [84], rats [85] and macaques [129]
VEGFR2, α_v_β_3_ integrin and P-selectin	Anti-VEGFR2, anti- α_v_β_3_ integrin and anti-P-selectin antibodies	Breast cancer in mice [139,140]

**Table 3 nanomaterials-10-01935-t003:** Summary of extravascular targets investigated for molecular US imaging.

Targets	Binding Ligands	Model
HER2	HER2-affibody	Breast cancer in mice [152,153]
	Anti-HER2 antibody	Breast cancer in mice [154]
CAIX	Aptamer	Adenocarcinoma in mice [7]
PSMA	PSMA-1 ligand	Prostate cancer in mice [151]
CD3	Anti-CD3 antibody	T-lymphocyte in rats [155]
CA-125	Anti-CA-125 antibody	Epithelial ovarian cancer in mice [156]
proGRP	Anti- proGRP antibody	Small cell lung cancer in mice [149]
Phosphatidylserine	Annexin V	Apoptosis in mice [157]
VEGFR2 and HER2	Anti-HER2 and anti-VEGFR2 antibodies	Breast cancer in mice [158]

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
