# Peer review of "Molecular Ultrasound Imaging"

_nanomaterials, 2020, doi:10.3390/nano10101935_

Round 1
Reviewer 1 Report
The Authors summarized more recent developments in the field of molecular US imaging. The paper deserves to be published "as it is".
Please, spelling "SPAQ" on line 17 (abstract)
Author Response
The Authors summarized more recent developments in the field of molecular US imaging. The paper deserves to be published "as it is".
Please, spelling "SPAQ" on line 17 (abstract).
We highly appreciate the reviewer’s positive feedback. We replaced SPAQ in the abstract by “Sensitive Particle Acoustic Quantification”.
Reviewer 2 Report
This review article by Koese, Darguzyte and Kiessling was written well, with a good and in-depth introduction to the technology. This reviewer has the following concerns/suggestions.
- The review, especially in the section of targeted contrast agents is extremely cancer-focused. Please expand on inflammation and cardiovascular areas.
- In the thrombosis area, there have been a lot of publications with RDG coupled microbubbles, as well as other platelets and/or coagulation pathway biomarkers. Please include some of them.
- For VCAM-1 targeting agents, please include more recent inflammation and cardiovascular studies. Some examples are PMID: 31597443 and PMID:
- Include a small section on the use of contrast agents for theranostic approaches and their use in drug and/or genetic therapeutics delivery via ultrasound.
- For a review article on imaging, there is only one in vivo image, please include molecular ultrasound images for different targeted and in vivo models.
- The reviewer understands the need to include publications from founders of this field, however, almost 15% of the 166 citations are from the same author/group. Diversity is needed.
- Excessive 15 citations (#14-27) for the section on biotin-avidin conjugation. Unsure why they are all needed? All these citations were also from before 2008.
- Care is needed for the references, formatting is not performed properly with some citations missing authors, journal name, year and issue etc. Please double check.
Author Response
This review article by Koese, Darguzyte and Kiessling was written well, with a good and in-depth introduction to the technology. This reviewer has the following concerns/suggestions.
We are grateful for the feedback. We have taken into consideration the suggestions as follows.
1. The review, especially in the section of targeted contrast agents is extremely cancer-focused. Please expand on inflammation and cardiovascular areas.
We agree with the reviewer. We expanded the sections about cardiovascular and inflammation imaging (see answer to questions 2 and 3).
2. In the thrombosis area, there have been a lot of publications with RDG coupled microbubbles, as well as other platelets and/or coagulation pathway biomarkers. Please include some of them.
We looked into the literature and added new publications of targeted microbubbles using RGD and other markers for thrombosis imaging. The following references were added:
- Moccetti, F.; Brown, E.; Xie, A.; Packwood, W.; Qi, Y.; Ruggeri, Z.; Shentu, W.; Chen, J.; López, J.A.; Lindner, J.R. Myocardial Infarction Produces Sustained Proinflammatory Endothelial Activation in Remote Arteries. Journal of the American College of Cardiology 2018, 72, 1015–1026, doi:10.1016/j.jacc.2018.06.044.
- Wu, W.; Wang, Y.; Shen, S.; Wu, J.; Guo, S.; Su, L.; Hou, F.; Wang, Z.; Liao, Y.; Bin, J. In Vivo Ultrasound Molecular Imaging of Inflammatory Thrombosis in Arteries With Cyclic Arg-Gly-Asp–Modified Microbubbles Targeted to Glycoprotein IIb/IIIa. Investigative Radiology 2013, 48, 803–812, doi:10.1097/RLI.0b013e318298652d.
- Hu, G.; Liu, C.; Liao, Y.; Yang, L.; Huang, R.; Wu, J.; Xie, J.; Bundhoo, K.; Liu, Y.; Bin, J. Ultrasound molecular imaging of arterial thrombi with novel microbubbles modified by cyclic RGD in vitro and in vivo. Thromb Haemost 2012, 107, 172–183, doi:10.1160/TH10-11-0701.
3. For VCAM-1 targeting agents, please include more recent inflammation and cardiovascular studies. Some examples are PMID: 31597443 and PMID:
As suggested, we added recent studies using VCAM-1 targeted microbubbles. In detail, the following papers were added:
- Punjabi, M.; Xu, L.; Ochoa-Espinosa, A.; Kosareva, A.; Wolff, T.; Murtaja, A.; Broisat, A.; Devoogdt, N.; Kaufmann, B.A. Ultrasound Molecular Imaging of Atherosclerosis With Nanobodies: Translatable Microbubble Targeting Murine and Human VCAM (Vascular Cell Adhesion Molecule) 1. Arterioscler. Thromb. Vasc. Biol. 2019, 39, 2520–2530, doi:10.1161/ATVBAHA.119.313088.
- Moccetti, F.; Weinkauf, C.C.; Davidson, B.P.; Belcik, J.T.; Marinelli, E.R.; Unger, E.; Lindner, J.R. Ultrasound Molecular Imaging of Atherosclerosis Using Small-Peptide Targeting Ligands Against Endothelial Markers of Inflammation and Oxidative Stress. Ultrasound in Medicine & Biology 2018, 44, 1155–1163, doi:10.1016/j.ultrasmedbio.2018.01.001.
- Koczera, P.; Appold, L.; Shi, Y.; Liu, M.; Dasgupta, A.; Pathak, V.; Ojha, T.; Fokong, S.; Wu, Z.; van Zandvoort, M.; et al. PBCA-based Polymeric Microbubbles for Molecular Imaging and Drug Delivery. J Control Release 2017, 259, 128–135, doi:10.1016/j.jconrel.2017.03.006.
- Wang, S.; Unnikrishnan, S.; Herbst, E.B.; Klibanov, A.L.; Mauldin, F.W.; Hossack, J.A. Ultrasound Molecular Imaging of Inflammation in Mouse Abdominal Aorta. Invest Radiol 2017, 52, 499–506, doi:10.1097/RLI.0000000000000373.
4. Include a small section on the use of contrast agents for theranostic approaches and their use in drug and/or genetic therapeutics delivery via ultrasound.
Since there are already excellent recent review articles on therapeutic and theranostic approaches with ultrasound, we decided to focus solely on molecular imaging. However, we added a few sentences in the introduction section to highlight this emerging field and added according references.
5. For a review article on imaging, there is only one in vivo image, please include molecular ultrasound images for different targeted and in vivo models.
Thank you for the suggestion. We included two additional in vivo images (figure 6 and 9).
6. The reviewer understands the need to include publications from founders of this field, however, almost 15% of the 166 citations are from the same author/group. Diversity is needed.
We checked for diversity and now ensure that references from a single group do not dominate the article, i.e. have more than 10 % of the total reference number.
7. Excessive 15 citations (#14-27) for the section on biotin-avidin conjugation. Unsure why they are all needed? All these citations were also from before 2008.
As suggested, we removed the citations in biotin-avidin conjugation part.
8. Care is needed for the references, formatting is not performed properly with some citations missing authors, journal name, year and issue etc. Please double check.
We carefully revised the references and their formatting.
Reviewer 3 Report
This is a really impressive review article on the potential of molecular ultrasound imaging in translational medicine. The authors present an impressive overview of the current methods and applications in a variety of diseases. The article is very well written. I recommend publication with any need for revisions.
Author Response
This is a really impressive review article on the potential of molecular ultrasound imaging in translational medicine. The authors present an impressive overview of the current methods and applications in a variety of diseases. The article is very well written. I recommend publication with any need for revisions.
We are excited that the reviewer likes our manuscript, and we are thankful for the positive remarks.
Reviewer 4 Report
The manuscript is a very well written and comprehensive review targeting the different aspects of molecular ultrasound imaging, with clear descriptions of the imaging technique and of the several contrast agents that have been developed for this modality. Pros and cons for each differente ultrasound contrast agent have been described in detail, allowing the reader to understand the obtained results in a large variety of applications (angiogenesis, inflammation, atherosclerosis and thrombosis).
Minor:
Figure 2: some words in the figure are difficult to read: power modulation or summed response: please check spaces between words
Pag 3, line 94 (Figure 2 legend): "For tissue response near zero,.." should be corrected to "For tissue the response is near zero, .."
Pag 9, line 278: Section 3.2.5 should be 3.3.5
Author Response
The manuscript is a very well written and comprehensive review targeting the different aspects of molecular ultrasound imaging, with clear descriptions of the imaging technique and of the several contrast agents that have been developed for this modality. Pros and cons for each different ultrasound contrast agent have been described in detail, allowing the reader to understand the obtained results in a large variety of applications (angiogenesis, inflammation, atherosclerosis and thrombosis).
Minor:
Figure 2: some words in the figure are difficult to read: power modulation or summed response: please check spaces between words
Pag 3, line 94 (Figure 2 legend): "For tissue response near zero,.." should be corrected to "For tissue the response is near zero, .."
Pag 9, line 278: Section 3.2.5 should be 3.3.5
We are grateful for the positive feedback. We adjusted the manuscript according to the comments.